# Representation Learning for Event-based Visuomotor Policies

**Sai Vemprala**
Microsoft Research
Redmond, WA - 98052
savempra@microsoft.com

**Sami Mian** *
University of Pittsburgh
Pittsburgh, PA - 15260
sami.mian@pitt.edu

**Ashish Kapoor**
Microsoft Research
Redmond, WA - 98052
akapoor@microsoft.com

## Abstract

Event-based cameras are dynamic vision sensors that provide asynchronous measurements of changes in per-pixel brightness at a microsecond level. This makes them significantly faster than conventional frame-based cameras, and an appealing choice for high-speed navigation. While an interesting sensor modality, this asynchronously streamed event data poses a challenge for machine learning techniques that are more suited for frame-based data. In this paper, we present an event variational autoencoder and show that it is feasible to learn compact representations directly from asynchronous spatiotemporal event data. Furthermore, we show that such pretrained representations can be used for event-based reinforcement learning instead of end-to-end reward driven perception. We validate this framework of learning event-based visuomotor policies by applying it to an obstacle avoidance scenario in simulation. Compared to techniques that treat event data as images, we show that representations learnt from event streams result in faster policy training, adapt to different control capacities, and demonstrate a higher degree of robustness.

## 1 Introduction

Autonomous navigation, which is driven by a tight coupling between perception and action, is particularly challenging for fast, agile robots such as unmanned micro aerial vehicles (MAV) that are often deployed in cluttered and low altitude areas. For such reactive navigation applications such as obstacle avoidance, low sensor latency is the key to performing agile maneuvers successfully [1]. MAVs are also limited in their size and payload capacity, which constrains onboard sensor choices to small, low-power sensors, and the computational load of the processing algorithms to be minimal.

Modern computer vision and machine learning techniques for perception and navigation typically focus on analyzing data from conventional CMOS based cameras, in various modalities such as RGB images, depth maps etc. While these cameras provide high resolution data, the main drawback of these sensors is their speed, with most averaging output at a rate of 30-60 Hz. This makes such sensors unable to scale up to the perception data rate required by agile navigation.

Inspired by biological vision, neuromorphic engineering has resulted in a novel sensor known as the dynamic vision sensor, or an event-based camera [2]. These cameras detect and measure changes in log-luminance on a per-pixel basis, and return information about 'events' with a temporal resolution on the order of microseconds. Due to the increased sampling speed of these cameras and the minimal processing needed to parse the data, perception using event cameras can be much faster than traditional approaches. This can allow for faster control schemes to be used, as enough relevant environmental information can be collected quickly in order to make informed control choices. Moreover, the events are inherently generated by changes in brightness typically arising from motion. This makes event cameras natural motion detectors and a good fit for learning control policies.

---

*Work done while interning at Microsoft Research

35th Conference on Neural Information Processing Systems (NeurIPS 2021).

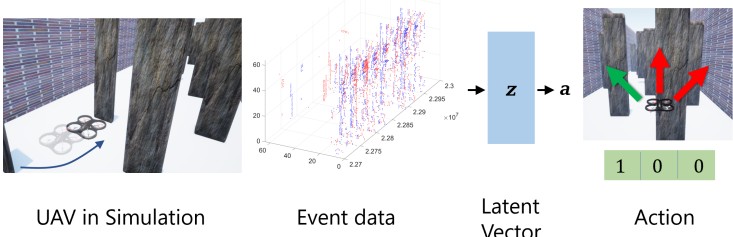

|  UAV in Simulation | Event data | Latent Vector | Action |

**Figure 1:** Event cameras provide fast, asynchronous measurements of per-pixel log luminance changes. We present a representation learning technique that can encode context from such spatiotemporal event bytestreams. Subsequently, we show these low-dimensional representations are beneficial for learning visuomotor policies through a simulated UAV obstacle avoidance task.

But the fundamentally different visual representation of event cameras poses significant challenges to quick adoption. Event cameras produce fast and asynchronous spatiotemporal data, significantly different from synchronous frame-based data expected by conventional machine learning algorithms. In addition, the quality of the data recorded by an event camera is different from traditional perception sensors; the sensors return low-level data that could vary significantly based on the firing order of pixels, lighting conditions, reflections or shadows.

Previous research has approached this modality through two main classes of techniques. Some approaches [3, 4] accumulate event data over time into a two dimensional frame, and use traditional computer vision/convolutional neural network based techniques with these frame-based inputs. Traditional CNN approaches combined with such accumulation fail to exploit the true advantages of event cameras such as the microsecond-scale temporal resolution, and may prove to be too intensive for high-speed action generation onboard constrained platforms. Another class of techniques involves the usage of spiking neural networks (SNN) [5]. SNNs operate through spiking neurons to identify spatio-temporal firings, making it a natural match for event cameras. Yet, training spiking neural networks is hard, as they do not use standard backpropagation, and often require specialized hardware to truly realize their efficiency [6, 7].

In this paper, we propose learning representations directly from raw event camera streams using conventional (non-spiking) machine learning methods, and learning visuomotor policies over such representations (Figure 1). We present an event variational autoencoder (eVAE) framework for learning representations from event data in a way that allows for high temporal resolution as well as invariance to data permutations and sparsity. The eVAE is equipped with an event feature computation network that can process asynchronous data from arbitrary sequence lengths, or in a recursive fashion. Inspired by the recent success of Transformer networks [8, 9, 10], the eVAE uses a temporal embedding method that preserves the event timing information when computing latent representations. Next, we show that such representations can be beneficial for reactive navigation, by applying them as observations in a reinforcement learning (RL) framework. We show how training RL policies over eVAE representations allows the control policy to generalize to different data rates and even to out-of-distribution environments. We define obstacle avoidance for UAVs as our task of interest and demonstrate how event camera data can be effectively utilized for avoidance at high control rates. Through an event data simulator, we simulate scenarios where the UAV is assumed to be controlled at up to 400 Hz, and show that the ability of the representations to handle sparse data allows the policy to adapt to high control rates. The key contributions of our work are listed below.

1. We present an event variational autoencoder for unsupervised representation learning from fast and asynchronous spatiotemporal event bytestream data.

2. We show that these event representations capture sufficient contextual information to be useful in learning reactive visuomotor policies.

3. We train policies over event representations using reinforcement learning for obstacle avoidance for UAVs in simulation and show that they outperform current state of the art in event-based reinforcement learning.

4. We discuss advantages of using bytestream representations for policies, such as: adaptation to different control capacities, robustness to environmental variations and noise.

## 2 Related Work

*Vision-based representations and navigation*: Variational autoencoders have been shown to be effective in learning well structured low-dimensional representations from complex visual data [11, 12, 13]. Leveraging such methods, recent research has focused on the decoupling of perception and planning, showing that separate networks for representation and navigation is effective [14, 15]. As the representation is expected to capture rich salient information about the world with a degree of invariance, this combination allows for higher sample efficiency and smaller policy network sizes [16].

*Feature learning from Event Cameras*: Some of the early work conducted on processing event data resulted in computing optical flow using the asynchronous data, focusing on high-speed computations with minimal bandwidth [17]. Event representations included histograms of averaged time surfaces (HATS), where temporal data is aggregated to create averaged data points capable of being used as input for traditional techniques [18] and hierarchy of event-based time surfaces (HOTS), another representation for pattern recognition [19].

*Learning from Sequences and Sets*: Learning from event data can be treated as a case of learning long, variable length sequences. While conventional RNNs are found to be infeasible for such lengths, approaches such as Phased LSTM [20] propose adding a time gate to LSTM for long sequences. If the spatial and temporal parts were decoupled, the problem can be reformulated as permutation-invariant learning from sets. Qi et al [21] present PointNet, which is a one such permutation invariant approach aimed at learning from 3D point cloud data. Similarly, Lee et al [22] present the Set Transformer, an attention-based learning method for sets.

*Event Cameras and Machine Learning*: From a machine learning perspective, Gehrig et al [23] introduced a full end-to-end pipeline for learning to represent event-based data, which discusses several variants such as event data aggregated into a grid-based representation, event spike tensors, and 3D voxel grids. Asynchronous versions of convolutional neural networks are also being developed to take advantage of the sparsity in data such as that of event cameras [24, 25]. Stacked spatial LSTM networks were used with event sequences for pose relocalization in [4]. EV-FlowNet [26] is an encoder-decoder architecture for self-supervised optical flow for events, which uses frame-based inputs processed through convolutional layers. The asynchronous nature of event data was handled through a permutation-invariant and recursive approach in EventNet [27]. Event camera based perception was used in other applications as well, such as self-supervised learning of optical flow [28], steering prediction for self driving cars [3]. Spiking neural networks were also used to examine event-based data [29, 30, 31, 32, 33, 34].

*Sensorimotor Policies with Event Cameras*: Only very recently has there been work on combining event camera data with sensorimotor policies. Event camera data was coupled with control for autonomous UAV landing in [35], [36]. EVDodge [37] creates an avoidance system for UAVs by using event data to track moving objects and infer safe avoidance maneuvers based on these measurements, combining multiple modules such as homography, segmentation, with the actions driven by a classical control policy. Event camera data was also used to power a closed-loop control scheme for a UAV on a fixed bench setup by tracking roll angles and angular velocities in [38]. Reinforcement learning using event camera data has only been explored very recently, using accumulated event frames fed into CNN-based policy networks for ground robots [39] and for UAV obstacle avoidance [40].

## 3 Representation Learning for Event Cameras

### 3.1 Event-based camera

An event based camera is a special vision sensor that measures changes in intensity levels independently at each of its pixels. Given a pixel location $(x, y)$, the fundamental working principle of an event-based camera is to measure the change in logarithmic brightness at that pixel, i.e., $\Delta log\, I(\{x, y\}, t)$ where $I$ is the photometric intensity. When this change in logarithmic brightness exceeds a set threshold, the camera generates an 'event', reporting the time and location of change, along with the 'sign' of the change. In contrast to conventional cameras which output a set number of frames per second, an event camera outputs events sparsely and asynchronously in time as a stream of bytes, which we refer to as an event 'bytestream'. These events are produced at a non-uniform

rate, and the number can range from zero to millions of events per second. For example, the DAVIS 240 camera [41] has a theoretical maximum limit of 12 million events per second.

## 3.2 Definitions and Notations

For an event camera of resolution $(H, W)$, an event can be defined as a tuple of four quantities $e = (t, x, y, p)$ where $t$ is a global timestamp at which the event was reported by the camera, $(x, y)$ the pixel coordinates, and $p$ the polarity. A sequence of events over a time window of $\tau$ can thus be represented as $E_\tau = \{e_i | t < i < t + \tau\}$. When sliding a constant time window of $\tau$ over a longer sequence of events, we can see that the length of $E_\tau$ will not be constant as the number of events fired in that interval would change based on environmental or sensory considerations. The events in $E_\tau$ can also be accumulated and represented as a corresponding event image frame $I_{E_\tau}$.

## 3.3 Event bytestream processing

Given event data as an arbitrarily long bytestream $E_\tau$, the objective of representation learning is to map it to a compressed vector representing the latent state of the environment $z_\tau$ through an encoder function $q_e(E_\tau)$. The challenges here are two-fold. First, due to the non-uniform and asynchronous nature of the event camera data, the same scene when imaged multiple times by an event camera could result in different permutations of the output. Hence, to handle the asynchronicity of event cameras, we require a feature computation technique that is invariant to data ordering. Secondly, while event sequences are time-based data, recurrent neural networks would prove to be infeasible due to the often long sequence lengths. Decoupling the temporal information from the spatial/polarity information alleviates this problem. We propose a backbone called the event context network (ECN) to achieve this for event data.

The ECN can be thought of as a preprocessing neural network for event streams, similar to architectures aimed at learning unordered spatial data such as PointNet [21] and EventNet [27]. The ECN takes an arbitrarily long list of events, and first computes a feature for each event. Eventually, these features are passed through a symmetric function (similar to PointNet, we also use a *max* operation), resulting in a global feature that is expected to condense information from all the events. The symmetric nature of this function ensures that these events in a given list can be processed either as a single batch, or recursively with any minibatch size to compute the output. We call the output of this feature network a 'context vector'. The ECN consists of three dense layers which, for $N$ input events, output an $N \times D$ set of features. The data passed into these dense layers is only the $(x, y, p)$ part of the events - and we discuss how we handle the temporal information next.

### 3.3.1 Temporal embedding

Timestamps in the event data inherently encode the continuous-time representation of the scene that was perceived during the given time slice, and it is important to retain them so the compressed representation is sufficiently informative of the evolution of the world state. On the other hand, incorporating the timestamp is not straightforward. Due to the asynchronicity of the data, a particular event may have any arbitrary timestamp within a given sequence. Hence, including the temporal data as an input to the ECN directly would interfere with the feature computation, as the global timestamps are arbitrary values, and even the relative time difference of each event would change every time new events are is received, necessitating a recomputation of the features.

Instead, we propose using 'temporal embeddings', inspired by the positional encoding principle that was first proposed for Transformer networks in [8]. For an event set $E_n$ with $n$ events, we first normalize the timestamps to $[0, 1]$ such that the timestamp corresponding to the end of the window maps to 1. This allows the model to encode recency of events in a generalizable way, thus allowing the model to understand which events are more important than the others, as they represent recent activity in the scene, regardless of the length of the event sequence. The ECN computes a d-dimensional temporal feature for each normalized timestamp as follows.

$$te(t, 2i) = sin\left(\frac{100t}{1000^{i/d}}\right), te(t, 2i + 1) = cos\left(\frac{100t}{1000^{i/d}}\right) \tag{1}$$

$$\text{where } i \in [0, d/2], t \in [0, 1]$$

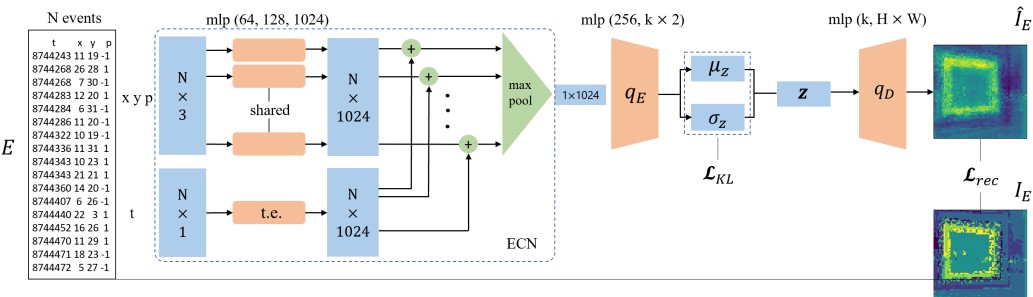

**Figure 2:** Architecture of the event variational autoencoder (eVAE). Events from the bytestream are directly processed by a PointNet-styled network to compute individual features. Temporal embeddings are added to these features and the *max* operation results in a global context vector. This is then projected into a latent space, and subsequently decoded into an 'event image'.

These embeddings are summed up with their corresponding features. The ECN passes this $N \times D$ feature set through the symmetric function *max* to obtain a $1 \times D$ final context vector. The ECN contains three dense layers for the feature computation along with the temporal embedding module and the max pool operator (Figure 2).

## 3.4 Event Variational Autoencoder

When learning representations for control, it is important for an efficient dimensionality reduction technique to create a smooth, continuous, and consistent representation. It is also desirable to have the encoded vectors' dimensions map to specific learned attributes of the perceived information, which can then be exploited by the control policies for interpretable learning. To achieve this, we extend the feature computation described in the previous section using variational autoencoders.

A variational autoencoder (VAE) [11] provides a probabilistic framework for mapping observations into a latent space. A VAE thus requires its encoder to describe a probability distribution for each latent attribute, instead of mapping attributes to outputs randomly. In the current framework, the event VAE (eVAE) operates on the context vector computed by the ECN and attempts to project it into a compressed latent space. Our encoder is composed of two dense layers as seen in Figure 2. In the decoding phase, instead of trying to reconstruct the entire input stream, we use an 'event image decoder' which attempts to decode the latent vector back to an approximate event image corresponding to the input sequence. This event image is a single channel image frame that is the result of accumulating all the events according to their pixel locations and polarity values, scaled by the relative timestamps. Similar to a standard VAE loss, the eVAE attempts to learn a parametric latent variable model by maximizing the marginal log-likelihood of the training data, composed of the event image reconstruction loss and a KL-divergence loss. The combined loss can be expressed as follows where the first term denotes the reconstruction loss, $P(z)$ is the distribution over the latent variable and $Q(z|x)$ is the approximated posterior by the VAE.

$$l(\theta) \geq \sum_{i=1}^{M} \mathbb{E}_{Q_i(z_i)}[log p_\theta(x_i|z_i)] - D_{KL}(Q_i(z_i|x_i)||p(z_i)) \tag{2}$$

A key thing to note here is that this deviates from the conventional definition of an autoencoder, where perfect reconstruction of input is sought. Instead, the eVAE's encoder-decoder structure operates upon the 'context vector', or the event features; and not the input stream itself. Hence, the goal is to encode the essence of the environment in a generalizable fashion, thus utilizing the low-level nature of the data. The decoder $q_D$ is another two dense-layer network that takes the (sampled) latent vector $z_\tau$ and outputs a reconstructed image $\hat{I}_{E\tau}$.

The training is performed end-to-end, so the weights for the ECN and encoder-decoder are all learnt simultaneously. While training, the eVAE can receive input data in two ways. The data can be passed as a set of batches with a predefined number of events per batch, or can be sliced according to a predefined time window where each window has a different number of events. During inference, as

in our application, the eVAE is expected to drive control commands, the length of the time window corresponds to the control frequency of the vehicle. This allows the context vectors to be computed either once at the end of the time window, or recursively at a faster rate where the context is computed and updated internally, and mapped to the latent vector when the control command is needed. More details about the eVAE training, computational effort etc. can be found in Appendices A and C.

## 4 Event-based Reinforcement Learning

Next, we focus on using event-based representations for navigation/planning purposes. While a straightforward approach would be to learn perceptual features together with actions, this would not scale well to event streams. As event cameras return data at a very high rate, relying on slow, sparse rewards to learn features in an end-to-end manner would be a disadvantage. Recent research has identified that generally, decoupling perception and policy networks and using intermediate representations enables faster training, higher performance and generalization ability [42]. We adapt this approach to event cameras, and propose using the eVAE representations in a reactive navigation framework. We define our task as collision avoidance for a quadrotor drone: where in simulation, the drone is expected to navigate from a start region to a goal region through an obstacle course, while avoiding collisions with any obstacle. Regardless of global positions of the drone or the obstacle(s), the drone should move in a particular direction that allows it to continue in collision-free areas, and repeat this behavior till the drone reaches its goal state. Hence, navigation and obstacle avoidance constitute a sequential decision making problem, which we address through reinforcement learning.

### 4.1 Background

We follow a conventional RL problem formulation for the reactive navigation task. As the quadrotor navigates in the environment and obtains event camera data, we pass the sequences output by the camera through the eVAE's encoder and consider the output latent vector $z$ to be the observation of the world state, such that $z_t = \mathcal{O}(.|s_t)$. The objective of the reinforcement learning approach is to learn a good policy $\pi_\theta(a|z)$.

We train our policies using the Proximal Policy Optimization (PPO) [43] algorithm. PPO is an on-policy policy gradient method, a class of methods that generally seek to compute an estimator of the policy gradient and use a stochastic gradient ascent algorithm over the network weights. The core principle of PPO is to 'clip' the extent of policy updates in order to avoid disastrously large changes to the policy. At time $t$, for an advantage function $A_t$ and for a given ratio of probability under new and old policies $r_t$, PPO solves a modified objective function for the estimator that can be written as:

$$L_{PG}^{clip}(\theta) = \hat{\mathbb{E}}\left[min(r_t(\theta)\hat{A}_t, clip_{1-\epsilon}^{1+\epsilon}(r_t(\theta))\hat{A}_t)\right] \tag{3}$$

### 4.2 Implementation

We create an obstacle avoidance scenario within the high fidelity quadrotor simulator AirSim [44], where a quadrotor drone is assumed to be equipped with a forward-facing event camera. We use an event simulator using the logarithmic image difference event generation model simulate events. To generate events across a span of time, it is necessary to first capture two images and compute the difference. Particularly when a high control frequency is desired (i.e., events should be computed and processed at a high rate), this complicates real time operation of the task. Due to this limitation, we instead use a steppable simulation. To emulate different control frequencies, we assume that the drone is moving at a constant predefined velocity and vary the step size of the actions dependent on the desired frequency. We assume the drone to be a simplistic model capable of moving at a speed of 20 m/s; thus, for example, the step size for a 200 Hz control would be 0.1 m. Further details about the RL training procedure and the environment can be found in Appendix D. In Figure 3, we show the environments used for training and testing the policies. In the interest of furthering research, we open source our representation learning and reinforcement learning framework along with the environments [2].

For the evaluation, we use a total of four policies, with two policies using the eVAE representations and two baselines.

---

[2]Our code and environments can be found at `https://github.com/microsoft/event-vae-rl`

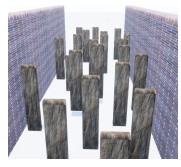 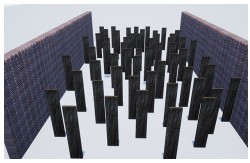 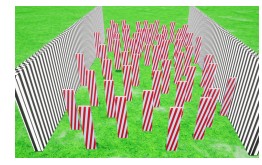 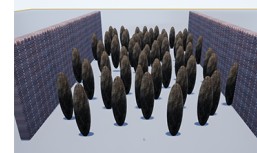

**(a)** RL Training environment  **(b)** RL Test environment  **(c)** RL Test - Texture change  **(d)** RL Test - Shape change

**Figure 3:** Environments used for RL training/evaluation

- *BRP-xy*: Policy learnt over pretrained eVAE representations encoding only XY locations from event data.

- *BRP-full*: Policy learnt over pretrained eVAE representations encoding full event data with timestamps and polarities.

- *EIP*: A policy with a CNN trained end-to-end, using the event image as input (similar to [39, 40])

- *EIVAE*: Policy learnt over a pretrained VAE that was trained using event images and a CNN backbone.

# 5 Results and Discussion

## 5.1 Representation Learning

Our first set of experiments aims to validate the learning of compressed representations encoded from the event sequences, and analyze the context-capturing ability of the eVAE. To train the eVAE, we simulate event data through AirSim's event simulator in three environments named *poles*, *cones*, and *gates* (drone racing gates), each indicative of the object of interest in it. More details about these environments can be found in Appendix B. The simulated event camera is assumed to be of $64 \times 64$ resolution and the data is collected by navigating in 2D around the objects.

*Qualitative performance*: Figure 4a displays the general performance of the eVAE at learning context out of event bytestreams. From the reconstructions, we observe that the eVAE latent space is able to encode the underlying essence of the input bytestream: locations of the objects, patterns of polarities, and information regarding the time of firing (brighter pixels in original/reconstruction indicate recent firings) are captured. We note that by encoding the arrangement of polarities, the latent space implicitly captures direction of motion, which in this case is due to the egomotion of the vehicle as we assume the environments to be static. In Appendix F, we show through a qualitative comparison that our proposed temporal embeddings result in better representation learning than the temporal coding approach proposed for event data in [27].

*Invariance to sparsity*: A key feature of the eVAE is its generalization to varying lengths of an event sequence, as the number of events at the input to the network can vary greatly. In Figure 4b, we show a comparison of the decoded image, when the eVAE is given sequences of different lengths starting at the same timestamp. The eVAE is quickly able to represent the object as a 'gate' once a minimum number of events matching that spatial arrangement are seen, and this projection into the latent space stays constant as more events are accumulated. Because the eVAE operates upon the context that is extracted from the stream, even short sequences are mapped to informative parts of the latent space based on the locations of the events. We compare this with a VAE trained on event images using a CNN encoder (Figure 4c), whose reconstruction quality degrades with decreasing sequence length, indicating the CNN's difficulty at handling sparse images.

*Generalization*: This context capturing ability also extends to unseen appearances of obstacles, highlighting the advantages of using low-level event data. In Figure 4e, we show samples of an eVAE trained on the *poles* data trying to decode data from the *cones* environment, and vice versa. Main environmental features (location of object, polarities etc.) are still captured by the latent vector, while the decoded image maps to the objects the eVAE has seen during training. This creates a degree of robustness in the eVAE specifically for reactive navigation: where the goal is to avoid obstacles no

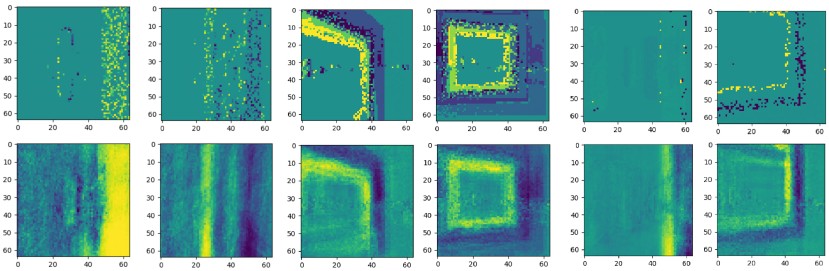

**(a)** Comparison of expected event frames (top) and reconstructed (bottom). The eVAE encodes locations of the obstacles and motion information from input sequences.

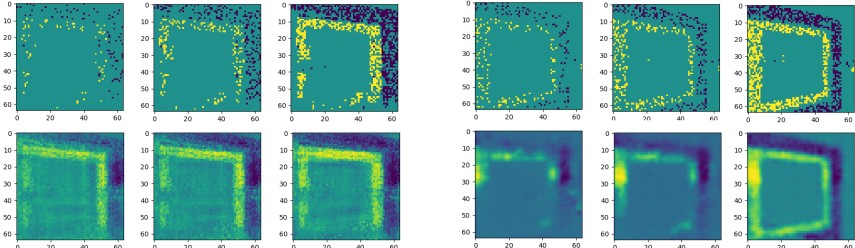

**(b)** eVAE reconstructions stay similar with changing data lengths (100, 200, 500 events).

**(c)** EIVAE reconstructions degrade for shorter lengths of event sequences.

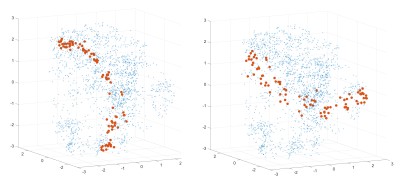

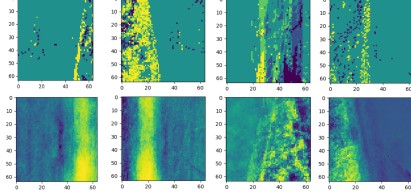

**(d)** Latent space transitions corresponding to the actions from a control policy form a structured trajectory in the space.

**(e)** eVAE representations when applied to out-of-distribution data, decode to seen objects while preserving context.

**Figure 4:** Qualitative results of the event variational autoencoder learning from various types of sequences.

matter what their shape/appearance is. We show in the later sections that this allows policies to work on radically different obstacle appearances without needing to retrain the policy.

*Smoothness of latent space*: As the eVAE combines the inherent manifold smoothness advantage of VAEs with high frequency input data, we observe that the smoothness automatically arises within the latent space as similar environmental factors map to the same latent variable values. We show an example in Figure 4d where we take a representation trained on the *gates* environment, which contains a set of drone racing gates, and observe the latent vectors when a drone navigates through the gates while collecting event observations. As the drone executes this set of actions, we see that the eVAE-encoded representation also shows a certain amount of structure. This way, state information from event data can potentially be projected into an approximately locally linear latent space, which has been shown to benefit high speed optimal control [45].

## 5.2 Reinforcement Learning for Obstacle Avoidance

*Policy training and control performance:* Next, we evaluate the results of using these pretrained representations as observations in a reinforcement learning framework for collision avoidance.

Considering that the bytestream-based policies are being trained over well structured lower dimensional representations, we observe improved performance during training. Comparison of the training rewards over the first 500000 timesteps can be seen in Figure 5a, where the bytestream representation policy (*BRP*) training is seen to have lower sample complexity than the event image policy (*EIP*)

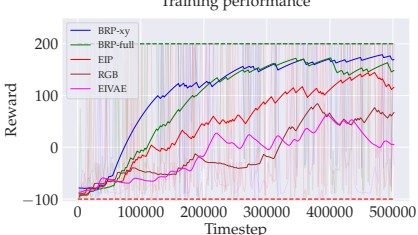

**(a)** eVAE representations allow faster training of policies compared to image-based policies.

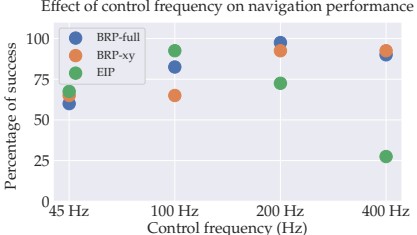

**(b)** eVAE representations benefit high speed control by being able to handle sparse sequences.

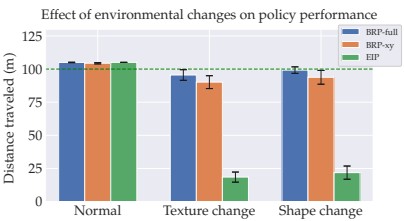

**(c)** eVAE based policies generalize better to obstacles of unseen textures or shapes.

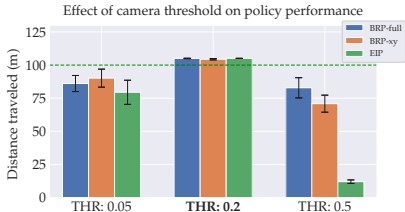

**(d)** eVAE based policies generalize to different event thresholds than seen in training.

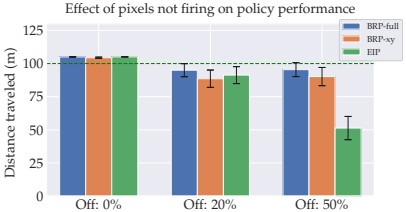

**(e)** Policies trained over eVAE representations exhibit robustness to induced sparsity in event data.

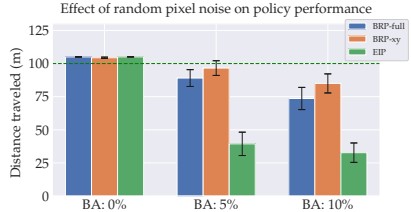

**(f)** Policies trained over eVAE representations exhibit robustness to additional noisy event firing.

**Figure 5:** Analysis of policy learning using event bytestream representations, compared against end-to-end trained event image + CNN policy.

or RGB images. We also find that the *EIVAE* policy it results in worse performance, which we hypothesize is due to its lower generalization ability to changing sequence lengths (e.g. very sparse images), which causes the latent representations to vary greatly for sequences that are too short or too long, making it harder to learn the corresponding actions.

Given the high data rate from event cameras, it is possible to control the vehicles at a higher frequency than with standard RGB camera images. We conduct an experiment where different control frequencies are simulated for the drone (varying the step size as discussed in section 4), and the trained *BRP* and *EIP* policies are tested. As conventional CMOS cameras often output data around 30-60 Hz, we choose 45 Hz as the minimum for the test, and 400 Hz (motor level control frequency of quadrotors) as the maximum. The results are seen in Figure 5b as success percentage over 40 trials in two environments, with success defined as whether or not the drone navigates through a 100m long obstacle course without collisions. We observe that all modalities suffer from low rate of success at 45 Hz, demonstrating the drawbacks of slow control in densely populated obstacle courses. At higher control frequencies, the motion of the camera and subsequently the number of events tend to be smaller. Even in these cases, similar to the observation in figure Figure 4b, extracting a latent representation allows the *BRP*s to be accurate, maintaining a high policy success rate at a simulated data rate of even 400 Hz. Intuitively, being able to perceive and control faster also means that the agent has enough chances to recover even in case of the occasional bad action. In contrast, we notice a falloff in the accuracy of *EIP* at higher control frequencies, as the event images get much sparser, which could prove to be problematic for a CNN.

*Robustness to environmental changes:* In the context of reactive navigation, the idea is to be able to avoid any obstacle regardless of characteristics like shape, appearance, texture etc. Through the *BRP*, we observe a key strength of the eVAE representations which is the generalization ability. First, we evaluate the performance when transferring a policy trained on the *poles* environment to unseen environments: one involving a change in texture of the obstacles, and another involving a change in shape (Figure 3). From the results in Figure 5c, we see that the *EIP* exhibits good performance on the environment it was trained on, but fails when applied to other environments due to the radically different obstacle appearances. Whereas, as seen in section 5.1, the eVAE brings a degree of invariance to the latent space projection, and hence the both *BRP*s perform better than the *EIP* with differently textured/shaped obstacles. We analyze this by running 20 trials of a policy under the test settings, and comparing the mean and standard error of the distance traveled without collision. In Appendix F, we show an additional experiment with a more complex out-of-distribution environment with obstacles that are of both different shape and texture containing moving textures, where the *BRP*s still maintain better performance.

*Robustness to camera parameters:* Similarly, we examine the effect of event camera sensor parameters on policy performance. For instance, in Figure 5d we examine the effect of the event threshold: which is the parameter that determines at what level of intensity change should an event be fired. A low value of threshold thus means a large number of events are fired, making the camera more sensitive to motion. When tested with different camera thresholds, which results in changing amounts of detail in the sequences, *BRP*s outperform the *EIP*. The eVAE affords the policies a degree of invariance to this redundant/unnecessary data, whereas the end-to-end CNN policy does not.

We also observe the bytestream representations benefiting the policy in case of induced sparsity in the event data. For this, we manually 'turn off', i.e., skip certain pixel locations in the event data. Figure 5e shows that the bytestream representation helps the policy maintain accuracy even up to the case where the event data is 50% sparser. Finally, event cameras are also prone to background activity (BA) [46], i.e., events being fired when there is no real intensity change. To simulate this, we add random events to the sequences. We observe that the *BRP*s still outperform the *EIP* (Figure 5f) - but we note that the BRPs are more sensitive to this type of noise than induced sparsity. In case of BA noise, *BRP-full* exhibits lower performance than *BRP-xy*, likely due to spurious polarities.

## 6    Conclusions

The event-based camera, being a low-level modality with fast data generation rate, is a good choice for high speed reactive behavior. Instead of treating event data as images, we present an event variational autoencoder that combines a spatiotemporal feature computation framework with the inherent advantages of variational autoencoders, enabling the learning of smooth and consistent representations directly from asynchronous event streams. By applying these representations in a reinforcement learning pipeline for navigation, we show that these representations effectively encode environmental context from fast streams, and can extract object locations, timing and motion information from polarity etc. in a way that generalizes over different sequence lengths and different types of objects, outperforming event image based methods.

**Limitations and Future Work.**   We present this work as an initial exploration towards connecting representation and reinforcement learning for event cameras. We have not tested this approach on large, diverse datasets, and some changes might be required to adapt these representations for more general representation learning with complex scenes. Our RL problem also focuses on a simpler setting due to computational issues with event simulation because complex scenes create more rendering overhead in simulators. Interesting avenues for future work could be leveraging GPUs for event simulation, applying these methods to harder tasks such as drone racing, and investigating recent advances in asynchronous convolutional networks [24] for representation learning.

**Broader Impact.**   On one hand, event cameras bring in a lot of potential for fast perception-action loops, such as for instance, integrating high speed reactive control with slow deliberative perception (thinking fast and slow [47]) for better robot intelligence. The low-level nature of event data also makes it a generally interesting candidate for vision, particularly for inducing shape bias as opposed to texture bias that is commonly seen in CNNs [48], and for privacy-preserving vision. On the other hand, there is potential for misuse with event cameras being used for surveillance, or drones equipped with those used in unethical applications, which requires careful consideration.

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
