# A    eVAE Details

## A.1    Network Architecture

Here, we describe the architecture of the eVAE presented in Figure 1 of the main paper, in more detail.

*Event Context Network*: We adapt the architecture proposed in [21] for the event context network, but without the feature transformation preprocessing steps. In our implementation, we use three Conv1d layers of 64, 128 and 1024 channels each followed by BatchNorm and a ReLU activation. At the end of the ECN, we add the temporal features (see Appendix A.2) to the $N \times 1024$ feature tensor, and execute the max operation to result in a context vector. The sizes of the intermediate features and the context feature are hyperparameters that can be varied based on the application, data complexity etc.

*Encoder*: The encoder for the VAE is composed of two layers, of sizes 1024 and 256 respectively, resulting in two output vectors of $1 \times 8$ each, corresponding to the mean and standard deviation for the latent space vector.

*Decoder*: The decoder for the VAE is composed of two layers, of sizes 256 and 1024 respectively, resulting in an output vector of size 4096, as our event camera image is supposed to be of $64 \times 64$ resolution. The final activation is Tanh if the polarity needs to be captured in the decoded image, otherwise, a sigmoid activation.

We use two variants of the eVAE: one that only learns from the XY pixel locations, and one that uses the full event data including polarity and timestamp. Subsequently, these two variants result in two policies: *BRP-full* and *BRP-xy*, as seen in Section 5 of the main paper.

## A.2    Temporal embedding

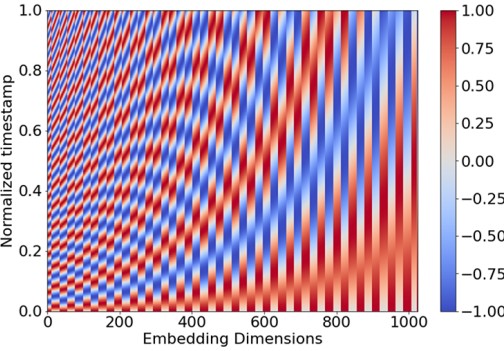

**Figure 6:** Visualization of $1 \times 1024$ temporal feature used as embedding, inspired by positional encoding in Transformers. [8]

Figure 6 shows a visualization of how the temporal features, computed by the temporal embedding equations vary over a window of time. Similar to the positional encoding in Transformers, they encode the normalized timestamp as a combination of sine/cosine values, which is then embedded into the event feature, and is subsequently decoded into a value that's indicative of the recency of the event. The set of parameters driving the temporal embedding equations (Eq. 1 in the main paper) were empirically chosen, based on qualitative results on the VAE validation set.

## Comparison with EventNet

EventNet [27] first proposed a way of temporal coding for event data such that the timestamp does not have to be part of the main feature computation, which would interfere with recursive processing. In EventNet, the normalized timestamp is proposed to be mapped into a complex phase function

representing a complex rotation. The equation from [27] for temporal coding modifies the feature as follows

$$a_{j,i} = c(z_i, \Delta t_{j,i}) = \left[ |z_i| - \frac{\Delta t_{j,i}}{\tau} \right] \exp\left( -i\frac{2\pi \Delta t_{j,i}}{\tau} \right) \tag{4}$$

This equation combines two modifications based on the normalized timestamp $t_{j,i}$ over a predetermined window $\tau$ - a linear decay of the feature magnitude, combined with a complex rotation. Due to the existence of a complex number, the max operation is defined as a complex max: i.e., a max over the magnitudes of the complex numbers involved, which would convert the complex phase into a magnitude of 1; thus the temporal factor would primarily decay the actual feature computed inside the context vector. EventNet uses a Tanh function to ensure the norm of the feature stays under 1, which means for older features in the window whose normalized timestamp is close to $\tau$, the feature value decays to close to zero.

In our goal of using event streams for reactive navigation, we are interested in reconstructing event data context from sparse and dense event streams equally importantly - and often, we would have to deal with sequences that could be dominated by 'old' events, in which case it is important to preserve those feature values as computed by the event context network. Hence, we propose using an additive temporal embedding that embeds the temporal information through a sinusoidal sequence. We conduct one experiment where we use EventNet's temporal coding system in the eVAE. We observe that it works well with larger event sequences, but when the event streams are sparser, or mainly contain older events inside a time window, the features fail to capture enough information possibly due to the decay. We show a qualitative comparison of EventNet's temporal coding scheme with ours in Figure 12, where we use the reconstructed event image as a measure of quality of how much information was captured in the latent space.

### A.3 Pseudocode

---

1   Encode
2   let $t_s = t$ #record sequence start
3   let $c = zeros()$
4   **while** $t < t_S + t_C$ **do**
5      $E = []$
6      $t' = t$
7      **while** $t < t' + t_F$ **do**
8          $e_{in} = (t, x, y, p)$
9          $e_{in}.t = (t - t_s)/(t_c)$ #normalize timestamp
10         $E$.append($e_i n$)
11      **end**
12      $c_{new} = ECN(E)$ #compute features and temporal embedding
13      $c = max(c, c_{new})$
14   **end**
15   $z = q_e(c)$ #compute latent vector
16   return $z$

---

During inference, when the eVAE is expected to compute latent vectors for streaming event data for use with control, we consider design choices for this computation to be driven by two main factors: the control frequency of the agent $f_C$, and a feature computation frequency $f_F$. The control frequency is the frequency at which the latent vectors are required to be computed, to be subsequently used for generating the actions. This is the minimum frequency the eVAE model has to adhere to. The feature computation frequency $f_F$ is a value such that $f_F \geq f_C$, which drives how often the context vector (the $1 \times D$ vector after max pooling) should be computed. In case the value of $f_C$ is low, which leaves long windows of time between control commands, it might be more optimal to compute $f_F$ more frequently, thus processing less number of events per batch and recursively update the context vector. Due to the recursive operation, similar to [27], the context vector can be computed even on a per-event basis - but we consider this to be both computationally infeasible and generally not required, considering events could be generated on the order of microseconds.

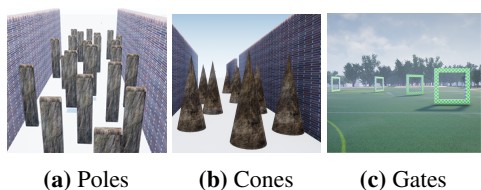

| **(a)** Poles | **(b)** Cones | **(c)** Gates |

**Figure 7:** Environments used for data generation for eVAE training

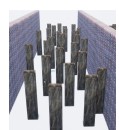 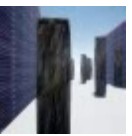 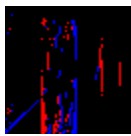

**Figure 8:** Training environment for obstacle avoidance, and a sample RGB image with corresponding event image view.

# B    eVAE Training Details

## B.1    Training Data

We generate event data by running an event camera simulator within Microsoft AirSim [44]. AirSim acts as a plugin for Unreal Engine, an AAA videogame engine that allows for simulating realistic graphics. AirSim provides models for quadrotors and cars along with a Python based API for robot control as well as gathering data such as RGB images. In this work, we use simplistic worlds to collect training data and obstacle avoidance environment. For the eVAE training data, We create three sample environments within AirSim, named *poles, cones, gates*. We first collect RGB camera data by moving a camera in two dimensions around this objects, which is subsequently converted to event bytestream data through an event simulator (see Appendix D). Pictures of these environments, and the objects within them can be seen in Figure 7.

## B.2    Procedure

We use batched data as input to the eVAE during training. We use a batch size of 2000 events, with 50 batches. The batches are selected by sampling randomly for an index from the full the event sequence, and then reading the subsequent number of events equal to the chosen batch size. We use the Adam optimizer with a learning rate of 0.001 and train for 20000 iterations.

We also use an annealing trick similar to the one proposed in [49] to avoid KL vanishing problems. To achieve this, we use the following scheduling cycle to impose a changing weight on the KL loss which is also appropriately weighted according to the values of reconstruction loss.

- $it < 1000$: $\beta = 0$
- $it > 1000$: $\beta = $ 1e-3 $* (it/10000)$

*Hyperparameters*: Learning rate - 0.001

# C    eVAE Computational Effort

In this section, we highlight a particular advantage of the event variational autoencoder. As seen in Figure 2, the eVAE encoder contains two major parts: the ECN and the subsequent two-layer MLP to result in a latent vector. In the ECN, which forms the main part of the encoder, we can see that the input values are always bounded. For example, the pixel and polarity values in the input data are discrete and bounded based on camera resolution $(W, H)$: $(x, y) \in Z : (x \in [0, W), y \in [0, H), p \in \{-1, 1\}$. This important characteristic means that the ECN only ever has to deal with a finite set of combinations as input, which allows a trained ECN to be reduced into an efficient lookup table to generate the output feature. We use $1 \times 1024$ as the feature size in our experiments, but this dimensionality can also be treated as a hyperparameter. Similarly, the temporal embeddings can also be precomputed, as

the timestamps within a window are also bounded to $[0, 1]$ over which a simple sin/cos encoding is applied. Hence, the encoder can be implemented, for example, as an efficient hash table with $\mathcal{O}(1)$ complexity followed by the 2-layer MLP responsible for the final latent vector. This fairly minimal amount of computation allows for deploying the eVAE encoder onboard constrained platforms such as micro aerial vehicles.

# D   Reinforcement Learning

## D.1   Training Environment

We use the poles environment to train the RL policies. The environment is approximately 100m in length, with randomly laid out poles as the obstacles, and two walls bordering it on the left and right. The lane is approximately 30m wide. The drone is spawned at one end of the lane, with an X coordinate ranging between $[-10, 10]$, creating a 20m wide feasible area for the start spot. The task is to navigate successfully through the course without any collisions. Reaching 100m+ in the Y direction counts as a success.

### D.1.1   Rewards

The rewards driving the RL policy training are as follows:

- +1 for each 1 m traveled in the direction of the goal (Y-axis)
- -100 for any collision, episode terminated.
- +100 for reaching 100m without collisions, episode terminated.

### D.1.2   Policies

We attempt to train a total of five policies using this framework. Two policies (*BRP-full* and *BRP-xy*) use the bytestream representation from the eVAE as the observation, whereas the third (*EIP*) uses an event image as observation. Naturally, the third policy uses a CNN to process the event image, and is trained end-to-end.

*BRP-full* is a policy that uses an eVAE latent vector as the observation; with the trained eVAE encoder processing the full event data of $(t, x, y, p)$. The latent vector is of shape $1 \times 8$, and a stack of three latent vectors is used as the observation.

*BRP-xy* is a policy that uses an eVAE latent vector as the observation; with the trained eVAE encoder processing only the $(x, y)$ part of the event data. The latent vector is of shape $1 \times 8$, and a stack of three latent vectors is used as the observation.

*EIP* uses a $64 \times 64 \times 3$ observation, with three most recent event image frames stacked into a single observation. This event image is an accumulation of all event data into a $64 \times 64$ frame with positive polarities represented as a pixel value of 255, and negative polarities as 125. Rest of the pixels stay 0. Temporal information is ignored in this event image. The *EIP* training loosely follows the implementation in [40].

Additionally, we also train a policy only using RGB images (RGB), and one using a VAE over the event images (EIVAE). We do not use them in our final comparisons and evaluations, as the RGB policy would naturally not be a generalizable modality, and the EIVAE was unable to reach acceptable performance, which we hypothesize to be because the VAE over images with CNN encoder and decoder is unable to properly map event images corresponding to sparse sequences as well as it does denser sequences.

*Policy networks*:

For policies *BRP-full* and *BRP-xy*, we use the standard MlpPolicy network from the stable-baselines framework [50]. This contains two dense layers of 64 neurons each.

For policy *EIP*, we use the standard CnnPolicy network from the stable-baselines framework, which borrows from [51].

The bytestream policies use a stack of the three most recent latent $z$ vectors as the observation ($1 \times 24$), whereas the event image policy uses a stack of the three most recent event images as the

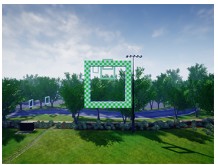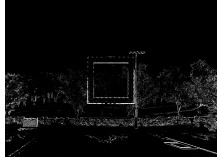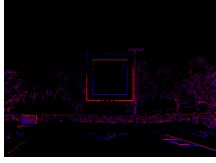

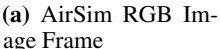

**(a)** AirSim RGB Image Frame  **(b)** Event Simulator output  **(c)** Event Simulator output with polarity visualization

observation ($3 \times 64 \times 64$). The *EIVAE* policy operates over a pretrained VAE representation that compresses the $3 \times 64 \times 64$ event image into a latent vector using the same CNN backbone as *EIP*.

*Hyperparameters*: We use the standard PPO implementation from stable baselines 3. While training, we use 2048 policy steps per update for a drone step size of 0.1m (corresponding to 200 Hz control). Other PPO hyperparameters are as follows: learning rate=0.0003, steps=2048, batch size=32, no. of epochs=10, gamma=0.99, gae lambda=0.95, clip range=0.2.

## D.2 Environments

The policies (both bytestream and image) are trained in the poles environment, using the trained weights from the eVAE encoder also corresponding to the poles environment.

We use a collection of environments for testing and evaluation. For more straightforward evaluation, we use two environments also containing poles as the main obstacles, but one containing a much denser layout, and the other a random arrangement with greater width. In order to evaluate the generalization ability of the policies as seen in results, we also create two new environments with new obstacles - one with the same texture, but different shape (ellipsoid), another with the same pole shape but different textures for the walls, poles and the ground. Pictures of these environments can be seen in Figure 3.

## D.3 Experiment details

*Control capacity analysis*: For this experiment, policies trained at a step size of 0.1m (corresponding to 200 Hz) were evaluated at step sizes corresponding to 45 Hz, 100 Hz, 200 Hz and 400 Hz. We run 20 trials in *poles-dense* and 20 trials in *poles-eval*, and report the percentage rate of success.

*Environmental changes*: For this experiment, policies trained in the 'poles' environment, and using the eVAE also trained on poles for the bytestream policies, were evaluated on environments with different texture and different shape. We run 20 trials each in *poles-eval, shape-change* and *text-change*.

*Camera threshold*: For this experiment, the policies were trained with a threshold of 0.2 in the event simulator. During evaluation, they were tested with threshold values of 0.05 and 0.2. We run 20 trials in *poles-eval* and report the average Y axis distance travelled by the drone without collision.

*Induced sparsity*: For this experiment, a certain percentage of pixels in the camera were assumed to be off - corresponding to these locations, any events that are reported by the simulator were ignored. The experiment was conducted with percentages of 0% (all pixels active), 20% and 50%. We run 20 trials in *poles-eval* and report the average Y axis distance travelled by the drone without collision.

*Background Activity*: Background activity (BA) is a type of noise in the event camera which triggers reporting of events without proper corresponding activity. For this experiment, the percentage of 'noise' pixels were chosen as a percentage of the total events reported (instead of percentage of the entire number of pixels) in order to avoid noise dominating the sequence too much. We choose percentages of 0% (no noise), 5% and 10% to conduct this experiment. We run 20 trials in *poles-eval* and report the average Y axis distance travelled by the drone without collision.

# E Event Camera Simulator

## E.1 General Operation

The event camera simulator used for this research was based on the simulator used with AirSim in [40], with several improvements made to increase the capabilities and accuracy of the output of the sim. The event simulator is now available as an open-source addon to AirSim [3]. The simulator works by comparing two frames of RGB video, calculating the difference in logarithmic intensity for each pixel over a time threshold, and comparing the results with a threshold. The algorithm below shows the steps to generate the output stream using pseudocode:

1. Take the difference between the log intensities of the current and previous frames

$$\Delta L(u,t) = log(I_t) - log(I_{t-1}) \tag{5}$$

2. Calculate the polarity for each each pixel based on a threshold of change in log intensity THR.

$$p(u,t) = \begin{cases} +1, & \text{if} \Delta L(u,t) > \text{THR} \\ -1, & \text{if} \Delta L(u,t) < \text{THR} \end{cases} \tag{6}$$

3. Determine the number of events to be fired per pixel, based on extent of intensity change over the threshold. Let $N_{max}$ be the maximum number of events that can occur at a single pixel

$$N_s(u,t) = min\left(N_{max}, \frac{\Delta L(u,t)}{\text{THR}}\right) \tag{7}$$

4. Determine the timestamps for each interpolated event

$$t = t_o + \frac{\Delta T}{N_s(u)} \tag{8}$$

5. Generate the output bytestream and sort by time.

In order to match the correct output of an event camera, in the final step the simulated events are sorted based on their temporal order. This way, all events that happened at the same time appear next to each other in the stream output. the resulting output is then written to an array for processing in our encoder, as well as saved to a text file if requested. Figure 9a shows the RGB input from AirSim, while Figure 9b and Figure 9c show the 2D reconstruction generated by the simulator. We implement this simulator in Python, and use optimization libraries such as Numba to accelerate performance.

The choice of this algorithm, and factors such as our relatively low resolution ($64 \times 64$) of the camera were to allow the event simulation, as well as AirSim image rendering to run as fast as possible to allow RL training/evaluation in near-realtime.

# F Misc. Results

## F.1 Dynamic textures

As an extension to the generalization experiments shown in section 5 for all three policies - two bytestream/one end-to-end event image; we conduct a final experiment with an even more complex setting: where the shapes of the obstacles are different, but the texture is also dynamic, i.e., moving (see Figure 10a). Due to this motion of the texture on the obstacles themselves, the polarity arrangements will be continuously changing. When we evaluate the three policies on this environment, we notice *BRP-xy* to perform the best, as it estimates environmental context (such as obstacle location) without considering the polarity of the events, but *BRP-full* degrades slightly, likely due to out of distribution polarity patterns. Both policies still demonstrate a large gain in performance over the end-to-end event image policy. The results of this experiment can be seen in Figure 10b. The latent space projections and subsequent decoding from out of distribution sequences (such as obstacles of different shapes/textures) is shown in further detail in Figure 11a and Figure 11b.

---

[3]For more details, please see `https://microsoft.github.io/AirSim/event_sim/`

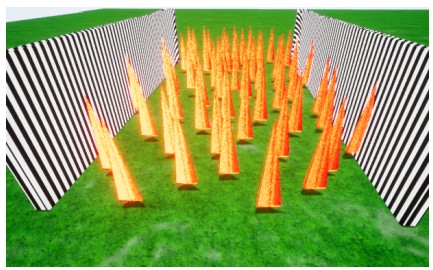 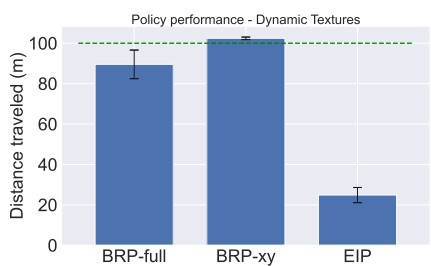

**(a)** Test environment with obstacles of both OOD texture as well as shape

**(b)** Average distance traveled without collisions in an environment with obstacles of OOD texture and shape

 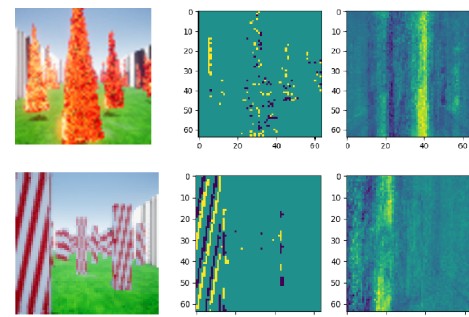

**(a)** Generalization to out of distribution appearances in *BRP-xy*. The decoded image indicates the latent space understands the obstacle position correctly.

**(b)** Generalization to out of distribution appearances in *BRP-full*. The decoded image indicates the latent space understands the obstacle position correctly.

### F.2 eVAE temporal embedding vs. EventNet phase temporal coding

See Figure 12 and Figure 13.



**Figure 12:** Qualitative comparison of eVAE latent space when using temporal coding (EventNet) vs. temporal embedding (proposed). Our method preserves the importance of older events, and thereby the decoder is able to output sharper reconstructions.

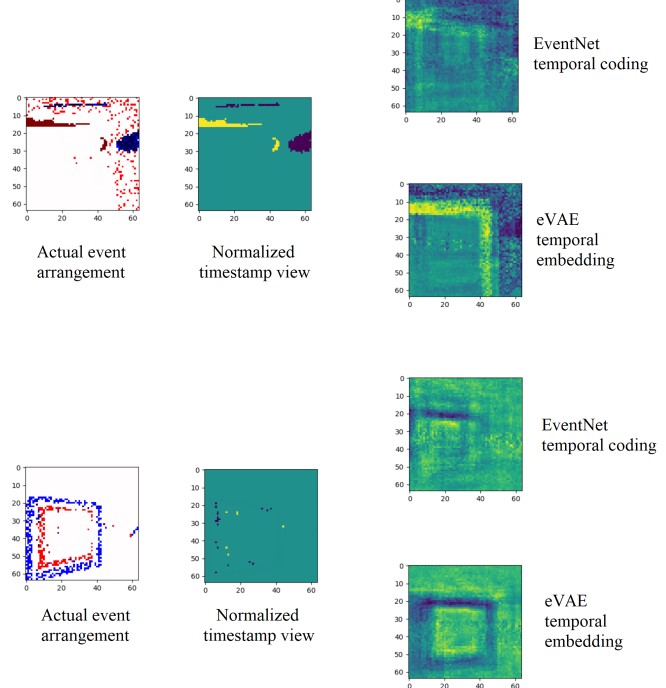

**Figure 13:** Qualitative comparison of eVAE latent space when using temporal coding (EventNet) vs. temporal embedding (proposed) - 2

 

   (b) Did you specify all the training details (e.g., data splits, hyperparameters, how they were chosen)? [Yes] See Appendix

   (c) Did you report error bars (e.g., with respect to the random seed after running experiments multiple times)? [Yes] See Figure 4

   (d) Did you include the total amount of compute and the type of resources used (e.g., type of GPUs, internal cluster, or cloud provider)? [Yes] See Appendix

4. If you are using existing assets (e.g., code, data, models) or curating/releasing new assets...

   (a) If your work uses existing assets, did you cite the creators? [Yes]

   (b) Did you mention the license of the assets? [N/A]

   (c) Did you include any new assets either in the supplemental material or as a URL? [Yes] We have released our simulation and RL train/test environments at `https://github.com/microsoft/event-vae-rl`.

   (d) Did you discuss whether and how consent was obtained from people whose data you're using/curating? [N/A]

   (e) Did you discuss whether the data you are using/curating contains personally identifiable information or offensive content? [N/A]

5. If you used crowdsourcing or conducted research with human subjects...

   (a) Did you include the full text of instructions given to participants and screenshots, if applicable? [N/A]

   (b) Did you describe any potential participant risks, with links to Institutional Review Board (IRB) approvals, if applicable? [N/A]

   (c) Did you include the estimated hourly wage paid to participants and the total amount spent on participant compensation? [N/A]