# OpenReview forum: "Representation Learning for Event-based Visuomotor Policies"
_NeurIPS.cc/2021/Conference — NeurIPS 2021 Spotlight_

### Official Review · Reviewer_guqw · 2021-07-14

**Rating:** 6
**Confidence:** 3

**Summary:**

This paper proposes a method for learning representations of event-based camera percepts which can be used for reinforcement learning. The method is able to digest irregular, streaming camera information using a VAE conditioned on a context vector computed from event data using a PointNet style network. The empirical evaluation shows that using this pretrained representation for PPO is more performant and generalizable than frame-based methods on a simulated UAV obstacle avoidance task.

**Limitations And Societal Impact:**

The method is tested only in one domain (simulated quadrotor flying, using the poles data). While the authors explain in Section 6 that this work is an initial exploration, it could make the paper much stronger to be able to demonstrate it with another task or more visually diverse scenes.

**Main Review:**

Originality:

The proposed eVAE method appears to be a novel approach to representation learning from this modality. The empirical results, particularly the generalization results, are strong compared to existing methods (of which there are not too many, as this is a relatively new direction of research).

Quality:

The experimental results provide reasonable support for the author’s claims, but are limited in scope by just one evaluation environment. The comparisons are appropriate and the proposed method achieves strong results.

My concerns in this section are the following:

The results presented in Fig 4A do not have error bars, could they be trained with multiple random seeds and added?

When training on one scene and testing on others, it seems like generalization occurs because unseen states are mapped to the closest possible latent. However, it’s not clear that this would result in a policy which actually reacts to the new obstacle shape. For example, if the cones were upside-down and had a significantly different shape as the poles, but got mapped to poles through the latent space, would the UAV be able to avoid them? This seems unlikely if trained using only one scene, but maybe if it were trained using a greater number of (or more diverse) environments, this would work?

Clarity:

Overall, the paper is well written. The authors have provided and documented code to make this work as reproducible as possible.
It could help improve clarity to address the following questions:

In Fig 4a, does the eVAE pretraining lead to improved asymptotic performance compared to the EIP policy? If the EIP policy is allowed to train for the same amount of effective gradient steps as the VAE training + RL training, will it achieve the same performance (or better, since it can learn features with the RL objective directly)?

It doesn’t appear that including the temporal or polarity information improves the performance of the learned policies. Do you have any explanation regarding why this information appears discardable for this task? Are there other tasks where it might not be?

Additionally, it could help to add some justification for the timestamps of a particular event window to be normalized to 1. Wouldn’t this cause event windows of differing lengths (in seconds) to become encoded to the same vector?

While there is some runtime analysis in the appendix, It would be interesting to compare the computational effort of the eVAE-based policy to the EIP-based policy in wall-clock time.

A brief description about how the event image is formed from the events would be helpful for understanding the VAE training process. How is the polarity involved in the computation and how is the relative time scaling performed?

Nitpicks:

The dotted line in Fig 4a is unlabeled. The actual values of rewards in the environment have been omitted from the main text, so it is unclear how to interpret them.

Significance:

As event based cameras could lead to the development of higher frequency control policies which are critical for certain types of tasks, the research question of how to best utilize this variable length asynchronous data stream is quite important.

Recommendation: Although the empirical evaluation could be more thorough in terms of tasks, environments, and more experimental trials, the eVAE method proposed is novel and appears effective, and seems like it will be quite impactful for future work using RL for control from event-based cameras. So, I think this is a good contribution and vote to accept.

**Time Spent Reviewing:**

3.5

---

> ### Author Response · Authors · 2021-08-10
> **Response to Reviewer guqw**
>
> We thank the reviewer for their feedback and helpful comments towards improving our paper. Please find our responses to the points in the review below:
>
> >The results presented in Fig 4A do not have error bars, could they be trained with multiple random seeds and added?
>
> We will try to add this in the final paper. Although figure 4(a) shows results from a single run, in our initial experiments, the general trend we noticed was that the BRPs train faster than the CNN-based policies.
>
> >When training on one scene and testing on others, it seems like generalization occurs because ...
>
> This is a great comment – because although the event VAE can figure out the approximate location and (relative) motion of an obstacle, by mapping unseen obstacles to known shapes, it no doubt loses some of the finer details. For instance, in the upside-down cone example, the UAV might end up hitting the top part of the cone as it underestimates the sizes of the obstacle. The solution to this would indeed be to train with more scenes (and perhaps increasing the latent space dimension to allow for richer representations), but in our results, we placed our focus on analyzing the generalization aspect of event-based representations and policies, as opposed to creating an obstacle avoidance policy that can be deployed universally.
>
> >In Fig 4a, does the eVAE pretraining lead to improved asymptotic performance compared to the EIP policy?
>
> In terms of general RL training performance, the EIP can indeed reach similar rewards over time as the BRP, although a bit slower. That figure was only meant to highlight the lower sample complexity of the BRPs as they benefit from the pretraining phase which simplifies the pixel space into a compressed representation space. But although one specific EIP can be trained well, we have observed in our experiments (and can be seen through the different test cases in figure 4), that the EIPs fail to generalize well to different test conditions such as environments, camera noise etc.
>
> >It doesn’t appear that including the temporal or polarity information improves the performance of the learned policies. Do you have any explanation regarding why this information appears discardable for this task?
>
> With the exception of 2-3 cases in figure 4, including the polarities does seem to help most of the time, although, granted, not by a lot. We attribute this to the simplicity of our task environments. The obstacles are static, and the UAV especially at higher control rates, has more samples to observe the obstacle even without the polarity information. Secondly, the input to the policy is a stack of 3 event representations. We believe polarities will become more critical when environments have fast-moving obstacles (UAV dodging obstacles for example), or perhaps when a single representation vector is provided as an input to the policy.
>
> >Additionally, it could help to add some justification for the timestamps of a particular event window to be normalized to 1. Wouldn’t this cause event windows of differing lengths (in seconds) to become encoded to the same vector?
>
> The intuition behind this was to allow the model to encode recency of events in a more general way. This allows the model to understand which events are 'more important' than the others, as they represent recent activity in the scene, regardless of the length of the event sequence. We note that this normalization is only for the timestamps, and hence will only affect the temporal embedding vector - the normal event context vector will be computed based on individual events through the ECN. In some (more complex) cases, one might require the model to encode finer details about the temporal aspect, and we wish to explore that in future work - perhaps by attempting to reconstruct the event stream itself rather than an approximated event image.
>
> >While there is some runtime analysis in the appendix, It would be interesting to compare the computational effort of the eVAE-based policy to the EIP-based policy in wall-clock time.
>
> As the BRP policy uses a simple MLP compared to the EIP, we see the BRP policy network itself to be much faster than the CNN: 12 us per forward pass for the BRP vs. 155 us for the CNN. But we note that using the BRP means computing the eVAE representation first – in this portion, we’ve observed an average of 1ms for the ECN and minimal time spent on the VAE encoder (< 50 us). Hence, with the proposal in appendix C that a trained MLP from the ECN can be replaced with a look-up table, the eVAE policy should see a gain in performance by largely minimizing the time spent in the ECN portion.
>
> Please note that these timing numbers were obtained with a desktop GPU, so they're merely for model-to-model comparison and not intended to reflect usage on real robots which often have lesser compute.
>
> >A brief description about how the event image is formed from the events would be helpful for understanding the VAE training process. How is the polarity involved in the computation and how is the relative time scaling performed?
>
> We will add more detail regarding this in the final paper. The event image is also formed through image differencing: and a value of +1 or -1 is obtained according to the polarity, and then multiplied with the relative time scale, so the resultant pixel is either dim (for old events) or bright (for new events). Some examples of such dim vs. bright events can be seen in Figure 12. The actual process of creating this event image can also be seen in the supplementary code: in the create_frame() function in event_vae/data_utils.py

---

### Official Review · Reviewer_sdni · 2021-07-17

**Rating:** 7
**Confidence:** 4

**Summary:**

The authors present an event-based variational auto-encoder (eVAE), which learns a latent space representation that is subsequently used for reinforcement learning (RL). The RL task consists of a simulated drone that has to avoid obstacles while flying forward in AirSim. Although the authors themselves do not mention this as a contribution in the paper, I think that the Event Context Network (ECN) they introduce is actually a major contribution of this article. In the ECN they combine an EventNet-like architecture with a Transformer-like position encoding (here temporal encoding). This ECN is a core part of the eVAE and works better than the original EventNet-way of encoding temporal order (as evidenced in the appendix). Finally, the authors show that with the eVAE, a learned policy transfers better to different environments.



**Ethical Concerns:**

No ethical concerns that were not identified by the authors.

**Limitations And Societal Impact:**

The authors have addressed both social impact and limitations of the work.

**Main Review:**

The proposed ECN and eVAE will be a valuable contribution to the construction of event-based latency spaces for - eventually - use on embedded robots. In terms of originality, the proposed method is primarily a very skillful combination of techniques found in the literature, but that is fine in my opinion.

The article is (apart from a few minor points) clear and well-written. However, I would advise the authors to include the new ECN as a contribution. This would also make the comparison with the original EventNet more important, which I would then suggest to also make part of the main article. I know that due to the page limits this may be challenging, but it could be added to, e.g., Fig 3 if the other panels there show 1 less example in order to make room for the EventNet comparison. On the topic of Figures, Fig 4 has super small legends, which are impossible to read.

The main criticism I would have on the article is related to the simplicity of the dataset (also mentioned by the authors under limitations). A very small latent space is learned (8-dimensional) and the examples shown by the authors that illustrate the superiority of the method over other methods mostly show that given a few events, the latent space will interpret an image still like the most likely pose of a pole or gate. The real world is obviously more complex and it would be interesting to see the eVAE in action for a real-world event-camera dataset (potentially with a larger latent space). This is even more relevant since the authors simulate events based on image differencing. Is there a study in the literature on how good this model is compared to the real outputs of an event-based camera?

This criticism extends to the used simulator and simulated environments, which is far cry from real autonomous drone flight. However, I do think it is valid as a first step, given the goals of the article. Moreover, as the authors mention themselves, event-based camera inputs will generalize better to different settings (including possibly the real world) in comparison to RGB values. The experiments in simulation already testify to this.

Smaller remarks:
- "Event camera data was also used to power a closed-loop control scheme for a UAV in flight by tracking roll angles and angular velocities in [38]." --> these experiments were not in flight, but with a fixed bench setup.
- For Eq 1 I had to go back to the Transformer paper. i and d are not (well) explained in the text, as only D was mentioned.
- Figure 2: how does the MLP at the start compare to the one in PointNet / EventNet (PointNet's is much more extensive I have the idea)
- Figure 3c: "eVAE representations stay consistent for different lengths of event sequences." What lengths? (I see three panels, but no indication of what the lengths were)
- "We note that by encoding the arrangement of polarities, the latent space implicitly captures direction of motion, " --> This is only true if you know the fore- & background color, right? Else there is ambiguity...
- Figure 4b: at 100Hz the EIP works best. Is there any explanation for this?
- The main article should have a picture of the simulated environment for the obstacle avoidance.

POST-REBUTTAL:

After reading all reviews & rebuttals, and the responses to my own comments, my score has not changed. If the authors perform the modifications they mention in the rebuttal, I think this will be a valuable contribution to the field.






**Time Spent Reviewing:**

6

---

> ### Author Response · Authors · 2021-08-10
> **Response to Reviewer sdni**
>
> We thank the reviewer for their feedback and insightful comments for improving the paper. Please find our responses below:
>
> >I would advise the authors to include the new ECN as a contribution. This would also make the comparison with the original EventNet more important, which I would then suggest to also make part of the main article.
>
> Thanks for this suggestion: we agree that this comparison could be well relevant to the main discussion. We will try to update the section to emphasize our contribution regarding the ECN and include some of the ECN vs. Eventnet comparison from the appendix within the results section.
>
> >The main criticism I would have on the article is related to the simplicity of the dataset (also mentioned by the authors under limitations).
>
> We acknowledge that our dataset and representations were fairly simple, but our focus in this work was to present an initial exploration towards learning representations from raw event data and how they can contribute towards robustness and generalization. As we can see from figure 4, even with a simple task/dataset, CNN based policies (or even a CNN-VAE one) had trouble generalizing to conditions different from training. Extending to real world data as well as more complex scenes in simulation are directions we definitely wish to pursue as future work.
> As to the image differencing model, most event simulation models currently out there are built upon image differencing such as eSim [1], CARLA [2] etc. A qualitative comparison of real vs. simulated events was provided by the eSim authors [3]. To account for some difference between real and simulated event cameras, we perform some tests with events manually turned off, or with spurious firing, attempting to mimic real noise (4(e), 4(f))
>
> [1] Henri Rebecq and Daniel Gehrig and Davide Scaramuzza, “ESIM: an Open Event Camera Simulator”.
>
> [2] https://carla.readthedocs.io/en/latest/ref_sensors/#dvs-camera
>
> [3] https://www.youtube.com/watch?v=VNWFkkTx4Ww
>
>
> Our responses to the minor remarks:
>
> >On the topic of Figures, Fig 4 has super small legends, which are impossible to read.
>
> We will update the figures to make the text clearer.
>
> >"Event camera data was also used to power a closed-loop control scheme for a UAV in flight by tracking roll angles and angular velocities in [38]." --> these experiments were not in flight, but with a fixed bench setup.
>
> This is true, thanks for pointing this out – we will amend our original sentence.
>
> >For Eq 1 I had to go back to the Transformer paper. i and d are not (well) explained in the text, as only D was mentioned.
>
> We will add a couple of lines explaining the terms and the significance of this equation in section 3.
>
> >Figure 2: how does the MLP at the start compare to the one in PointNet / EventNet (PointNet's is much more extensive I have the idea)
>
> Yes, PointNet is more extensive. The initial MLP in our ECN matches the ‘second half’ of the PointNet model. But the first half in PointNet deals with input transformation, and then a feature transformation. The input transformation makes sure that different point clouds in the training data are aligned by transforming them into a common space (e.g. centering around a common origin), whereas for events we don’t need that as they’re all in the same pixel space. The feature transformation then transforms these aligned Nx3 points into Nx64 features that are also realigned, but in the ECN, we skip this step.
>
> >Figure 3c: "eVAE representations stay consistent for different lengths of event sequences." What lengths? (I see three panels, but no indication of what the lengths were)
>
> The 3 figures in 3(c) correspond to event sequence lengths of 100, 200 and 500 events each. We will add this info to the caption.
>
> >"We note that by encoding the arrangement of polarities, the latent space implicitly captures direction of motion, " --> This is only true if you know the fore- & background color, right? Else there is ambiguity...
>
> This might differ a bit on a case-to-case basis, but generally, foreground objects generate more events than background ones because of difference in pixel motion. Hence our intuition was that if a foreground object was moving to the right, we would see many negative polarities to the left and a number of positive polarities to the right as the object occupies more pixels towards the right. Arrangements such as this could be captured by the representations.
>
> >Figure 4b: at 100Hz the EIP works best. Is there any explanation for this?
>
> The best performing EIP policy was trained with the actions being taken at 100 Hz. At 200 Hz, the event images became a bit too sparse, and the CNN based policy had trouble converging. We realize that we missed this point in the appendix (appendix says all policies were trained at 200 Hz) and will mention it.
>
> >The main article should have a picture of the simulated environment for the obstacle avoidance.
>
> We will add this to the main paper.

---

> > ### Comment · Reviewer_sdni · 2021-09-10
> > **Post-rebuttal**
> >
> > I have edited my review based on the reviews, rebuttals and discussion between reviewers. I am satisfied with the authors' rebuttal and the modifications they propose to make for the final version. Please see my POST-REBUTTAL remark in the edited review.

---

### Official Review · Reviewer_twzt · 2021-07-17

**Rating:** 6
**Confidence:** 3

**Summary:**

The author proposed an event based VAE for high speed robot navigation useful for reinforcement learning. They then compare to other image based RL approaches in AirSim and show they outperform other event based RL baselines. They demonstrate that the representation is useful for training visuomotor policies. They show that it reaches SOTA in event based reinforcement learning on the given tasks.

**Limitations And Societal Impact:**

Yes

**Main Review:**

# Originality
The paper proposed an event based VAE and uses it for downstream reinforcement learning. The paper focuses on using high sampling rate data streams which is an important and not very often studied problem in RL (particularly due to the amount it slows down training in simulators like Airsim.

# Quality
The paper provides a compelling argument that their technique performs better on high sampling rates particularly through the graphs in figure 4. They demonstrate that the paper reached SOTA on the obstacle avoidance task. I also appreciated the detailed discussion section explaining the benefits of the approach in regards to generalization. I also am glad that they examine the full event based representation and the simplified xy version and found different performance between the two. I don't see any issues with the methodology, but the RL problem is rather simplistic as the authors note. Especially given the small size of dataset and latent space, I am concerned about whether it would generalize to real world environments. However, this paper demonstrates that the method is more robust than RGB inputs, and therefore this is a promising first step.

# Clarity
I appreciate figure 3 for providing good qualitative visualization of the event based autoencoder, but I would recommend making the text larger in the final version. The impact of figure 4 is reduced due to the difficult of reading it. The paper text, itself is very well written and understandable. The supplemental material aided in the understanding of the paper as well so much so that parts of it (like an example of the environment should have been included in the main paper).

Would the authors clarify what they mean by: "Our RL problem also focuses on a simpler setting due to computational issues in event simulation" ? Is that saying that event based policies are computationally expensive? Or that the high framerate simulation required for training them is the main bottleneck? Specifying which computational issues are the constraints here, (speed, memory, gpu memory, physics simulation wall clock, graphics rendering, inference or training bottleneck in one of the network architectures etc.) is important for future authors who wish to build upon the approach.

The main issue in clarity is that the simulation environment is not shown qualitatively in the main paper which made the problem setup difficult to visualize.

# Significance
This representation could prove useful for drone in particular due to their high speed, but also is applicable to other high speed robotics. Additionally, the representation learned seems to have potential for improving sim2real transfer due to the robustness to shape and texture within simulation as shown in the appendix as well. They also demonstrate the importance of high frequency control in how all techniques struggle at a lower frequency of samples at 45hz.

I will admit that the paper does a good job proposing an interesting observation space that may generalize better for sim2real tasks as the paper demonstrates. With the changes to paper clarity and depending on the discussion of the computational bottlenecks, I would be willing to raise my score.

Edit: I would also like to apologize to the authors that my original review was truncated. I have edited my review.

Post Rebuttal:
Thank you for the authors detailed rebuttal. Ultimately, the rebuttal did not sufficiently address my concerns to raise my score beyond a 6.

**Time Spent Reviewing:**

4

---

> ### Author Response · Authors · 2021-08-06
> **Some text possibly missing from review**
>
> We thank the reviewer for their feedback and positive comments about the quality of our technique and presentation. It seems that some of the comments in the original review might have been missing: the 'quality' section ends with "I don't see any issues with the methodology, but here-", and Significance ends with "I will admit-". If the reviewer could please update their comments, we would be happy to respond to them by the time the official discussion period starts.

---

> ### Author Response · Authors · 2021-08-10
> **Response to Reviewer twzt**
>
> We thank the reviewer for their feedback and insightful comments. We are encouraged to see that our technique and presentation come across as clear and of significance. Please find our responses to the questions below.
>
> >To the question regarding computational issues - "… Is that saying that event based policies are computationally expensive? Or that the high framerate simulation required for training them is the main bottleneck?"
>
> It is the latter. Because event data encodes the change in intensities over time, ‘simulating’ event camera data in a conventional robot simulator involves capturing two images, taking the difference and then computing what events would have been fired within the time period between the two images. In photorealistic simulators such as AirSim, real-time image rendering is often computationally quite expensive for complex scenes, hence in our initial exploration, we limited ourselves to a simpler scene with a small image resolution. Secondly, realtime control with full dynamics is also challenging when simulating event data: let's say we were simulating a drone with agile dynamics, and we capture two images at times t1 and t2. At t2, we can begin computing th events corresponding to this time period through image differencing, but the drone would perhaps fly forward even as this computation is still being done. So we might have to reset the simulation to bring the drone back to t2 and then take the computed action. Event simulation is not much of an issue for tasks like classification/detection, which can operate on precomputed datasets, but can be more challenging for real-time control.
>
> That said, there are a few potential ways to improve this particular computational bottleneck – perhaps by leveraging GPU compute directly (as photorealistic simulators like AirSim build upon videogame engines such as Unreal Engine). Most event simulators today rely on simple per-pixel subtractions between two images to compute events (so does ours, along with a few optimizations). If event data simulation were written as a CUDA kernel, or a HLSL material shader, this could greatly improve the computational time. Secondly, for simulation, it might be more feasible to treat the control problem as an imitation learning problem, where we attempt to learn from state-action pairs containing event-representations and associated actions which does not require real-time exploration.
>
> Also as a response to reviewer guqw's comments, we've done a bit of wall clock time testing of the computational performance of our models which might be relevant here. Please note that these numbers were obtained with a desktop GPU, so they're merely for model-to-model comparison and not intended to reflect usage on real robots which often have lesser compute.
>
> BRP policy network forward pass: 12 us
> EIP policy network forward pass: 155 us
>
> But we note that using the BRP means computing the eVAE representation first – in this portion, we’ve observed an average of 1ms for the ECN and minimal time spent on the VAE encoder (< 50 us). Hence, with the proposal in appendix C that a trained MLP from the ECN can be replaced with a look-up table, the eVAE policy should see a gain in performance by largely minimizing the time spent in the ECN portion.
>
> >The main issue in clarity is that the simulation environment is not shown qualitatively in the main paper which made the problem setup difficult to visualize.
>
> We agree that this is indeed important info. We will include some pictures and details of the train/test environments in the main paper to help better visualize the problem setup; as well as the extent of generalization we were aiming for in the evaluation.

---

### Official Review · Reviewer_ZPhQ · 2021-07-23

**Rating:** 7
**Confidence:** 4

**Summary:**

The paper targets navigation and obstacle avoidance using high-frequency event cameras. A new event VAE (eVAE) is proposed to encode inputs from event cameras. An unsupervised learning scheme for eVAE is also proposed. The encoder is first evaluated qualitatively under variations of the input data. The pre-trained eVAE is then used to learn RL navigation policies. Results favorably compare to alternative encoding methods.

**Ethical Concerns:**

No specific issues other than standard for vision and robotics

**Limitations And Societal Impact:**

No specific issues other than standard for vision and robotics

**Main Review:**

Pros:
- Learning efficient visual representations for RL policies is an interesting and open problem
- Learning visiomotor policies for evet cameras should be interesting for many applications
- The paper is well-written and is easy to follow in general.
- The proposed eVAE encoding is evaluated and compared to baselines under variations of observed scenes and used parameters, e.g. length of event sequences
- The eVAE-based RL policy is tested for textures and shapes unseen during training and while changing the sparsity of event data
- Results are convincing given the comparison to baselines.

Cons:
- The definition of terms in eq.(2) is missing. The exact form of the loss for learning eVAE is hence unclear. What are Q,z,x in (2)?
- The robot seems to be always moving forward. What will happen if gets to a trap with no free passage for moving forward?
- Results for RBP-xy are similar to RBP. Does this mean that "e" part in eVAE is not important? Is the contribution of eVAE validated in this case?
- I have not found a commitment by the authors to release the experimental setup and environments. Such a release will be crucial for future comparisons.

**Time Spent Reviewing:**

4

---

> ### Author Response · Authors · 2021-08-10
> **Response to Reviewer ZPhQ**
>
> We thank the reviewer for their positive feedback about our paper and their helpful comments towards improvement. Please find our responses to the Cons section below:
>
> >The definition of terms in eq.(2) is missing. The exact form of the loss for learning eVAE is hence unclear. What are Q,z,x in (2)?
>
> We can add some more clarification at equation 2 to make it clearer. As it is, it is just representing the standard ELBO loss for a variational autoencoder where Q(z|x) indicates the approximated posterior by the VAE, and P(z) is the distribution over the latent variable. The event-VAE is a combination of the ECN and a standard VAE, so it is attempting to project the context vector output by the ECN into a compressed latent space. The loss for the eVAE thus is a combination of the reconstruction loss of the event image (MSE loss) and the standard KL divergence loss.
>
> >The robot seems to be always moving forward. What will happen if gets to a trap with no free passage for moving forward?
>
> In our current setting, we only focused on simpler obstacle courses, and limited our analysis to control frequency, robustness to appearance etc. We do not handle more complex motion such as turning around/better avoidance maneuvers, but we believe handling such cases is generally a matter of having more diverse training environments and/or a bigger action space. It is also possible to extend this representation learning framework to an imitation learning scenario, where one can learn richer action spaces from expert data.
>
> >Results for RBP-xy are similar to RBP. Does this mean that "e" part in eVAE is not important? Is the contribution of eVAE validated in this case?
>
> Across a majority of our tests, BRP-full (BRP is short for bytestream policies) does perform better than BRP-xy. Both the BRPs are policies trained over eVAE representations, so the event-VAE is a critical part of them – and the BRPs outperform the other CNN based techniques. Hence, it’s validated that handling event data through eVAE representations is more advantageous. We believe that the difference between BRP-full and BRP-xy could increase in more challenging domains such as fast moving obstacles, where the temporal information becomes more critical.
>
> >I have not found a commitment by the authors to release the experimental setup and environments. Such a release will be crucial for future comparisons.
>
> We will certainly release the environments as well as the code for the eVAE and policy training to benefit future research.

---

### Decision · Program_Chairs · 2021-09-27

**Decision:**

Accept (Spotlight)

**Comment:**

This paper introduces a new type of variational auto-encoders that can handle streams of high-frequency event-based spatiotemporal data (t, x, y, p) in order to decode dense images from a small set of N events, then combines a pre-trained eVAE with traditional reinforcement learning (PPO) to learn visuomotor policies to control (and fly while avoiding obstacles) a quadcopter in the AirSim simulator.
Reviewers have praised the idea, the application of RL to event-based data streams, the writing of the paper and the extent of experiment and comparison to non-event-based VAE + RL. Reviewers had some questions about equations, about specifics of some ablations, about rendering time for event-based simulation, about showing the simulator, etc., that were all satisfactorily answered by the authors.
After careful consideration of the paper and given review scores of (6, 6, 7, 7, mean 6.5), it appears that the paper should be accepted. The authors are invited to carry out all the changes they promised to the reviewers, including open-sourcing their code.